# Evaluation of Carbon Ion Radiation-Induced Trismus in Head and Neck Tumors Using Dose-Volume Histograms

**DOI:** 10.3390/cancers12113116

**Published:** 2020-10-25

**Authors:** Atsushi Musha, Hirofumi Shimada, Nobuteru Kubo, Hidemasa Kawamura, Naoko Okano, Yuhei Miyasaka, Hiro Sato, Katsuyuki Shirai, Jun-ichi Saitoh, Satoshi Yokoo, Kazuaki Chikamatsu, Tatsuya Ohno

**Affiliations:** 1Gunma University Heavy Ion Medical Center, Maebashi, Gunma 371-8511, Japan; shimada@gunma-u.ac.jp (H.S.); kubo@gunma-u.ac.jp (N.K.); kawa@gunma-u.ac.jp (H.K.); okano.n@gunma-u.ac.jp (N.O.); y.miyasaka@gunma-u.ac.jp (Y.M.); hiro.sato@gunma-u.ac.jp (H.S.); tohno@gunma-u.ac.jp (T.O.); 2Department of Oral and Maxillofacial Surgery/Plastic Surgery, Gunma University Graduate School of Medicine, Maebashi, Gunma 371-8511, Japan; syokoo@gunma-u.ac.jp; 3Department of Radiology, Jichi Medical University Hospital, Shimotsuke, Tochigi 329-0498, Japan; kshirai@jichi.ac.jp; 4Department of Radiation Oncology, Faculty of Medicine, University of Toyama, Toyama 930-0194, Japan; junsaito@med.u-toyama.ac.jp; 5Department of Otolaryngology-Head and Neck Surgery, Gunma University Graduate School of Medicine, Maebashi, Gunma 371-8511, Japan; tikamatu@gunma-u.ac.jp

**Keywords:** carbon ion radiotherapy, head and neck tumor, carbon ion radiotherapy-induced trismus

## Abstract

**Simple Summary:**

Patients who receive carbon ion radiotherapy (C-ion RT) for tumors near the temporomandibular joint are likely to experience trismus, a condition characterized by reduced jaw opening. However, the relationship between the dose of carbon ion and the onset of trismus remains unclear. Therefore, we conducted a subgroup analysis of a prospective observational study to understand the relationship between the dose of carbon ion and the occurrence of trismus. Of 35 patients included in the study, six developed grade 2 trismus, and the median onset time was 12 months. The affected muscles included masticatory muscles and the coronoid process. Our findings suggest better treatment planning, such as dose optimization, to minimize the occurrence of muscle-related adverse effects associated with C-ion RT.

**Abstract:**

Carbon ion radiotherapy (C-ion RT) provides a highly localized deposition of energy that can increase radiation doses to tumors while minimizing irradiation of adjacent normal tissues. For tumors located near the temporomandibular joint, C-ion RT-induced trismus may occur. However, the relationship between the carbon ion dose and the onset of trismus is unclear. In this prospective observational study, we assessed the trismus/carbon ion dose relationship using dose−volume histograms in 35 patients who received C-ion RT in their head and neck regions between 2010 and 2014. Trismus was evaluated in patients according to the Common Terminology Criteria for Adverse Events, version 4.0. All patients were treated with 57.6 or 64.0 Gy (relative biological effectiveness (RBE)) in 16 fractions, and the median follow-up time was 57 months. Grade 2 trismus was observed in six patients. The median onset time was 12 months. At maximum radiation doses, all masticatory muscles and coronoid processes, particularly the masseter muscle, were significantly different (*p* = 0.003). The contouring of the masseter muscle and coronoid process requires different treatment planning. The maximum radiation doses of the coronoid process can be proposed as a guideline for treatment planning, considering the ease of contouring in C-ion RT.

## 1. Introduction

Head and neck tumor patients undergoing radiotherapy suffer from acute adverse events (mucositis and dermatitis), and late adverse events (dysgeusia, osteoradionecrosis, and trismus). Usually it is challenging to improve late adverse events that reduce the patient’s quality of life (QOL). Radiation-induced trismus impacts the patient’s QOL, making it difficult for them to open their mouths, to eat, to talk, and to maintain oral hygiene [1,2]. The prevalence of X-ray-induced trismus for head and neck tumors is 1.4–41% [3,4,5,6,7,8,9,10,11]. Previous studies showed trismus onset by X-ray correlated significantly with the radiation dose received in the masseter muscles [1,2,6,10,12], pterygoid muscles [1,13,14], and in the temporomandibular joint [2,15], focusing efforts on reducing the radiation dose received by the above structures to mitigate the X-ray-induced onset of trismus. However, this is difficult when the tumor has invaded near the temporomandibular joint structures, increasing the concentration area of radiation-induced adverse events, since radiotherapy has become highly precise. In particular, a highly localized deposition by carbon ion radiotherapy (C-ion RT) energy can increase radiation doses to tumors while minimizing the irradiation area of adjacent, normal tissues. Previous studies have reported the incidence of trismus onset by C-ion RT. The prevalence of trismus by C-ion RT for head and neck tumors is 2–9% [16,17,18,19,20]. Therefore, the relationship between C-ion RT and trismus is unclear.

This study aimed to identify the correlation between the maximum dose and the dose−volume histogram, and C-ion RT-induced trismus.

## 2. Results

### 2.1. Incidence of C-Ion RT-Induced Trismus

The median follow-up time was 57 months. C-ion RT-induced trismus was evaluated using the Common Terminology Criteria for Adverse Events (CTCAE), version 4.0. [21], and grade 2 or higher was considered as trismus. Grade 2 trismus was observed in six patients (19.4%). There were no cases of grade 1 or 3 trismus. The median onset time was 12 months (range, 10–22 months). The trismus resolved entirely after 24 months of C-ion RT (Figure 1). Trismus onset showed no significant difference between the age, sex, primary site, histological type, T stage, and gross tumor volume (Appendix A).

### 2.2. Representative Images of a Tumor and Temporomandibular Joint-Related Muscles and Bones

Figure 2 shows a representative case of computed tomography images of the tumor and temporomandibular joint-related muscles and bones and a 3-dimensional image of the mandible. This patient had adenoid cystic carcinoma of the right maxillary sinus. The tumor and temporomandibular joint-related organs are close and irradiated with a high dose of radiation (Figure 2a). The masseter muscle, mandible head, and coronoid process were displayed on the 3-dimensional image of the mandible (Figure 2b). The high-dose region can be seen in front of the masseter muscle and the coronoid process (Figure 2b). In this patient, trismus was onset at 22 months after C-ion RT.

### 2.3. Dose−Volume Histograms and C-Ion RT-Induced Trismus

Figure 3 compares the dose−volume histogram (DVH) of the temporomandibular joint-related muscles and bones of patients with and without trismus. In the temporomandibular joint-related muscles, medial and lateral pterygoid muscle shared most of the high-dose area with or without trismus (Figure 3c,d). In contrast, the masseter muscle shared the least high-dose area with or without trismus (Figure 3a). The temporomandibular joint-related bones had different tendencies for DVH values with or without trismus (Figure 3e,f). In cases of trismus, the coronoid process tended to have a higher dose than the mandible head (Figure 3e,f).

### 2.4. Maximum Dose in the Temporomandibular Joint-Related Structures and C-Ion RT-Induced Trismus

The maximum radiation dose leading to no trismus or its onset was significantly different among various types of masticatory muscles (Table 1). In particular, the masseter muscle showed the most significant difference among its different muscles (Figure 4a, Table 1, *p* = 0.003). The maximum dose received by the masseter muscle that resulted in no trismus was 47.9 ± 19.0 Gy(RBE), whereas the maximum dose that caused trismus was 61.2 ± 5.9 Gy(RBE), which was significantly different. From the receiver operating characteristic (ROC) curve, the cut-off value was found to be 44.0 Gy(RBE) for trismus (sensitivity: 1.0, specificity: 0.44, AUC: 0.653, Table 1). In contrast, the difference between the maximum doses received by the bone structure of the temporomandibular that led to no trismus, 33.0 ± 20.8 Gy(RBE), or trismus, 54.8 ± 11.5 Gy(RBE), were significantly different in the coronoid process (Figure 4e, Table 1, *p* = 0.002) but not in the mandible head (Figure 4f, Table 1, *p* = 0.39). From the ROC curve, the cut-off value was found to be 38.0 Gy(RBE) for trismus (sensitivity, 1.0; specificity, 0.56; AUC, 0.773; Table 1). There were no cases in which trismus was absent when high doses of radiation were administered to both the temporomandibular joint-related muscles and bones.

### 2.5. Dose Rate of the Temporomandibular Joint-Related Structures and C-Ion RT-Induced Trismus

The doses received by 10, 20, 30, 40, and 50% (D10, D20, D30, D40, and D50) of the temporomandibular joint-related muscle and bone volumes, along with mean above structure dose in percentage are summarized in Table 2. The coronoid process showed significantly different doses for the presence or absence of trismus in all groups from D10 to D50 (D10; trismus 52.2 ± 13.0 Gy(RBE) without trismus, 29.3 ± 20.5 Gy(RBE) *p* = 0.007, D20; trismus 50.9 ± 13.3 Gy(RBE) without trismus, 27.8 ± 20.5 Gy(RBE) *p* = 0.007, D30; trismus 49.7 ± 13.5 Gy(RBE) without trismus, 26.7 ± 20.4 Gy(RBE) *p* = 0.007, D40; trismus 48.6 ± 13.6 Gy(RBE) without trismus 25.8 ± 20.2 Gy(RBE) *p* = 0.007, D50; trismus 47.4 ± 13.7 Gy(RBE) without trismus 25.0 ± 20.0 Gy(RBE) *p* = 0.008 ). At other sites, a significant difference was observed only in D10 of the temporal muscle (with trismus 50.3 ± 14.1 Gy(RBE) without trismus 34.2 ± 22.2 Gy(RBE), *p* = 0.024).

## 3. Discussion

In this study, we analyzed the maximum dose and DVH associated with trismus in head and neck cancer patients treated with C-ion RT. The median follow-up time was 57 months. Grade 2 trismus was observed in six patients. The prevalence of trismus was 19.4%, and the median onset time was 12 months. The prevalence of trismus induced by X-ray and C-ion RT is reported to be 1.4—41% [3,4,5,6,7,8,9,10,11] and 2–9% [16,17,18,19,20], respectively. In this study, the prevalence of a slightly higher incidence than previously reported may be due to the tumor site; however, this difference is not significant because this is a small study. A previous study showed that the onset of trismus is associated with a median of 1−16 months after completion of X-ray radiotherapy [2,4,8,9,10,12,15,22,23]; however, C-ion RT-induced trismus has not been reported. Most studies on X-ray-induced trismus have defined a mouth opening distance of less than 35 mm [2,6,8,9,10,23]. In contrast, as in this study, trismus in reporting C-ion RT is defined according to the CTCAE criteria [17,18,19,20,22].

Previous studies showed that trismus onset by X-ray had a significant correlation with the C-ion RT dose in the masseter muscles [1,2,6,10,12], pterygoid muscles [1,13,14], the temporomandibular joint [2,15]. The mean dose to the masseter muscle with trismus was 57.2 Gy at D50 [6]. In another report, after a dose of 40 Gy, for every additional 10 Gy radiation in the pterygoid muscle, an increase in the probability of trismus by 24% was observed [1]. However, in this study, the mean radiation doses administered to the masseter and pterygoid muscles were not significantly different for the onset of trismus at any time (D10, D20, D30, D40, and D50). Similarly, the mean radiation dose administered to muscles was not significant, except for the temporal muscle of D10; however, the radiation doses administered to muscles were significantly different for the maximum dose. In this report, a significant difference was confirmed, especially in the masseter muscle maximum dose, and the cut-off value was 44.0 Gy(RBE) for C-ion RT-induced trismus. Similar results have been reported with X-rays [1,2,6,10,12]. In contrast, in the coronoid process, there was a significant difference in both the mean dose (D10; 52.2 Gy(RBE), D20; 50.9 Gy(RBE), D30; 49.7i Gy(RBE), D40; 48.6 Gy(RBE), and D50; 47.4 Gy(RBE)) and the maximum dose (cut-off value 38.0 Gy(RBE)) associated with the onset of trismus.

In the DVH analysis, not only the high dose of radiation received by the masseter muscle but also the low to middle dose range received by the coronoid process seemed to be associated with the development of C-ion RT-induced trismus. Therefore, the reduction of the low to moderate dose volume of the DVH of the coronoid process may be useful in preventing trismus. The maximum dose administered to the coronoid process was also significant; however, it is unlikely that the high dose to the bone structure led to the trismus; the effect of radiation on the temporal muscle (at a muscular attachment), and the tendon may also be important. In some cases, high maximum doses were found even in non-onset cases, but the maximum dose was the maximum point dose, and due to the good dose distribution characteristic of C-ion RT, high-dose sites were spotted. Considering the ratio of radiation dose to each structure, it is possible that the middle to low dose area occupies most of the structure. According to the results in the present study, at the maximum doses of radiation, all types of masticatory muscles showed a significant difference in the development of trismus, with the most significant difference observed in the masseter muscle. We can offer a dose-constraint option in the C-ion RT optimization process, including a maximum dose to the masseter muscle of approximately 44.0 Gy(RBE). Moreover, from D10 to D50, it is important to keep the radiation dose to the coronoid process to less than 47 Gy(RBE). However, the contouring of the masseter muscle and the coronoid process requires different treatment planning for radiation oncologists (Figure 2b). Since the coronoid process forms contours more easily than the masseter muscle, radiation oncologists should consider it a risk organ to prevent C-ion RT-induced trismus.

As mentioned above, for non-invading tumors, minimizing radiation exposure to temporomandibular joint-related structures by reducing the dose of radiation therapy prevents the onset of trismus. However, this dose reduction strategy is challenging when the tumor invades temporomandibular joint-related structures. Several stretching techniques and jaw mobilizing devices are currently available to treat radiotherapy-induced trismus [24,25]. Jaw opening exercises using various jaw mobilizing devices, such as the TheraBite Jaw Motion Rehabilitation System and the Dynasplint Trismus System, have been proposed for treating radiotherapy-induced trismus [24,25]. There is, however, a lack of standard jaw mobilizing devices to treat radiotherapy-induced trismus. Our institution also recommends implementing mouth opening exercises using a device for patients who experience trismus within a year after C-ion RT. Future studies should consider the appropriate timing of mouth opening exercise, duration of mouth opening exercise, and type of device used to prevent radiation-induced trismus.

This study had a few limitations. First, since this study was conducted on a small number of patients enrolled at a single institute, the primary sites of tumor initiation were not examined thoroughly. Future studies should increase the number of patients for the study design. Second, we only evaluated the CTCAE criteria. In past studies using X-rays, many studies have defined a mouth opening of less than 35 mm as a mouth opening disorder [2,6,8,9,10,23]. As in a previous study using the CTCAE criteria [3], we focused on whether it was easy to open the mouth because there are individual differences in the mouth opening disorder.

## 4. Materials and Methods 

### 4.1. Patients and Tumor Characteristics 

The present study is a subgroup analysis of a prospective clinical study that included 35 patients diagnosed with nonsquamous cell carcinoma of the head and neck region and treated with C-ion RT between 2010 and 2014 in our institution. This study was approved by our Institutional Review Board and registered with the University Hospital Medical Information Network in Japan (trial registration number: UMIN000007886) [16]. All patients provided informed consent before treatment. For analyzing trismus, we excluded one case with previously existing C-ion RT-induced trismus, and three patients who underwent salvage surgery for recurrence after C-ion RT. Thirty-one cases were eventually analyzed (Figure 5). Table 3 summarizes the patient and tumor characteristics. The primary cancer sites were the maxillary sinus (n = 8), nasal cavity (n = 8), parotid gland (n = 5), oral cavity (n = 4), pharynx (n = 4), and the external auditory canal (n = 2). The 3-year local control rate (93%) and the 3-year overall survival rate (88%) for these patients have already been reported [16]. 

### 4.2. Carbon Ion Radiotherapy

The techniques used for C-ion RT and the treatment plan have been reported previously [16]. Physical dose calculations were performed using a pencil beam algorithm. The clinical dose distribution was calculated using the physical dose and relative biological effectiveness (RBE). The dose of C-ion RT was expressed as “Gy(RBE)” (physical carbon ion dose (Gy)×RBE). The number of fractions was 16, and the overall treatment time was four weeks (4 fractions per week). Following the clinical protocol, 29 patients received 64.0 Gy(RBE) in 16 fractions, 2 patients received 57.6 Gy(RBE) (in these two patients, the mucosa and skin were considered to be widely irradiated).

### 4.3. Analysis of Temporomandibular Joint Structures

In the muscle and bone structures of the diseased side, around the temporomandibular joint, the doses of radiation received by the masseter muscle, temporal muscle, medial pterygoid muscle, lateral pterygoid muscle, coronoid process, and mandibular head were examined. Using commercially available software (MIM Maestro, version 6.9.3, Beachwood, OH, USA), the contour of each organ and a 3-dimensional image of the mandible (Figure 2b) were created. Trismus was evaluated using the CTCAE, version 4.0. [21] and grade 2 or higher was considered to be trismus. The relationship between the dose of radiation and trismus was analyzed in various temporomandibular joint structures.

### 4.4. Statistical Analysis

Data are represented as mean ± standard deviation (S.D.). Statistical differences were compared using a two-sided Student’s t-test. A paired t-test was used to compare differences in the maximum doses between high-grade trismus and none. ROC curves were generated to anticipate the dose at the site of trismus. All data were analyzed using SPSS Statistics software, version 26.0 (IBM Corp., Armonk, NY, USA). Differences with *p* < 0.05 were considered statistically significant.

## 5. Conclusions

The masseter muscle showed the most significant difference between the presence and absence of trismus, in terms of maximum doses received among muscle tissues. The maximum and mean radiation doses that led to no trismus, or caused trismus, were significantly different in the coronoid process. The coronoid process can be suggested as a guideline for treatment planning considering the ease of contouring.

## Figures and Tables

**Figure 1 cancers-12-03116-f001:**
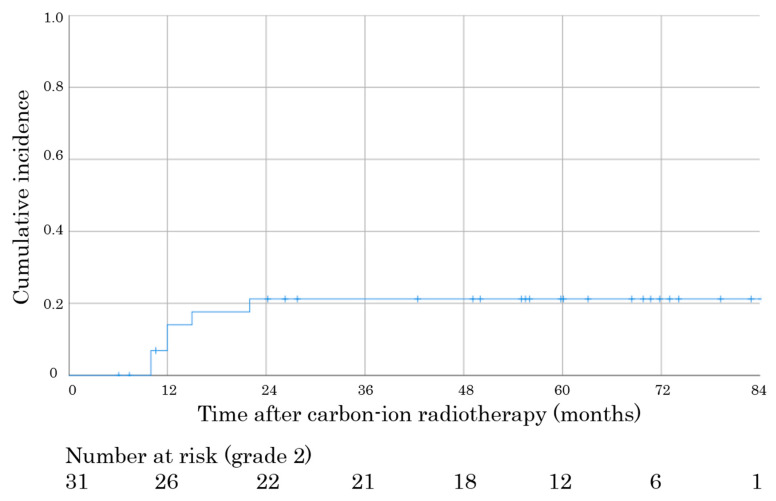
Cumulative incidence of grade 2 carbon ion radiotherapy-induced trismus after carbon ion radiotherapy in patients with this study (n = 31).

**Figure 2 cancers-12-03116-f002:**
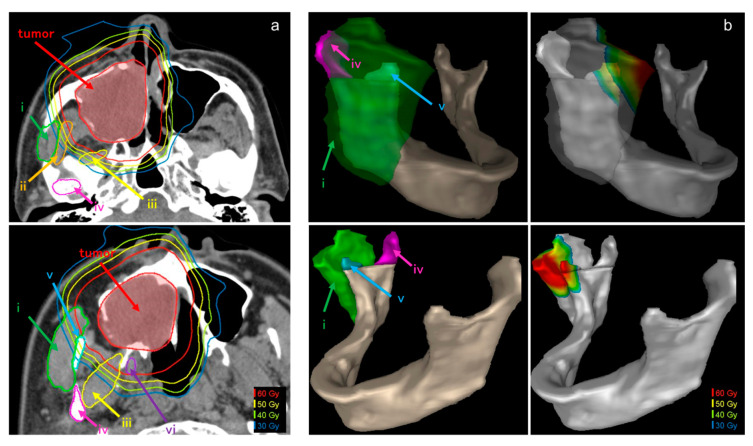
Representative case of computed tomography images of a tumor and temporomandibular joint-related muscles and bones and a 3-dimensional image of the mandible. (**a**) Computed tomography images of a tumor and temporomandibular joint-related muscles and bones. This patient with adenoid cystic carcinoma of the right maxillary sinus (T3N0M0). (**b**) The 3-dimensional image of the mandible with masseter muscle, mandible head, and coronoid process. The left figure shows the anatomical position. The figure on the right shows the dose distribution. (i) Masseter muscle, (ii) temporal muscle, (iii) lateral pterygoid muscle, (iv) mandible head, (v) coronoid process, and (vi) medial pterygoid muscle and tumor.

**Figure 3 cancers-12-03116-f003:**
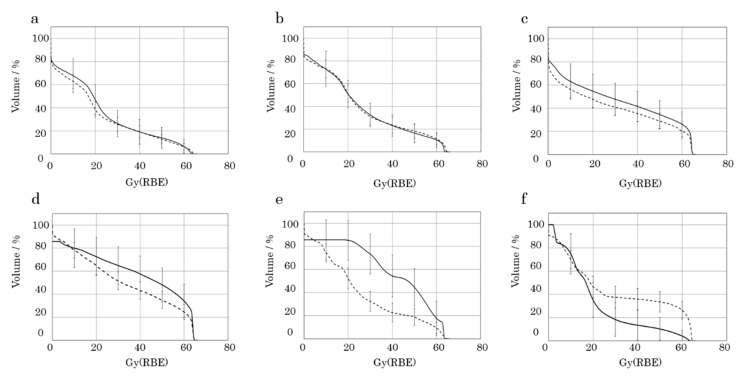
Comparison of dose−volume histograms and carbon ion radiotherapy-induced trismus. The vertical axis is the relative volume of the structure that received a higher dose than the absolute dose specified on the horizontal axis. (**a**) Masseter muscle, (**b**) temporal muscle, (**c**) medial pterygoid muscle, (**d**) lateral pterygoid muscle, (**e**) coronoid process, (**f**) mandible head. Continuous lines are with trismus, and dotted lines are without trismus.

**Figure 4 cancers-12-03116-f004:**
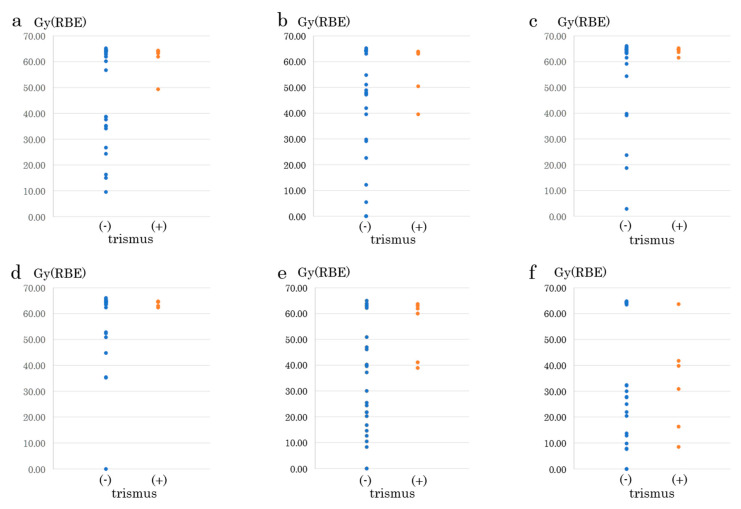
Relationship between maximum dose in the temporomandibular joint-related structures and carbon ion radiotherapy-induced trismus. Data are presented as mean ± S.D. (**a**) Masseter muscle, (**b**) temporal muscle, (**c**) medial pterygoid muscle, (**d**) lateral pterygoid muscle, (**e**) coronoid process, (**f**) mandible head.

**Figure 5 cancers-12-03116-f005:**
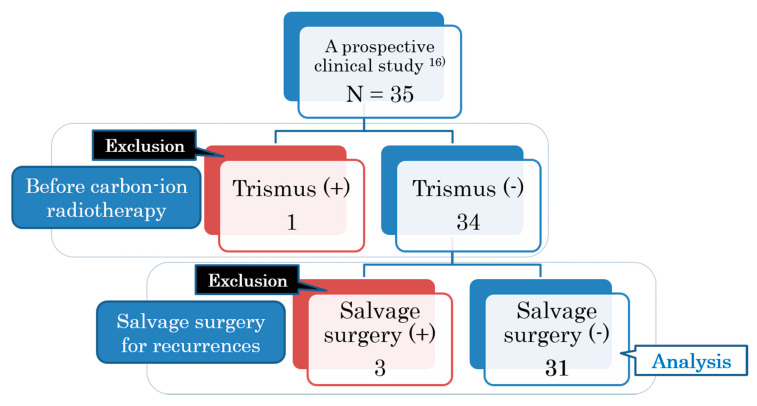
Patient selection criteria for the study.

**Table 1 cancers-12-03116-t001:** Cut-off value of maximum dose for the onset of trismus in temporomandibular joint-related organs.

Structure	Maximum Dose Cut-Off Value, Gy(RBE)	*p-*Value	Sensitivity	Specificity	AUC
Masseter muscle	44.0	0.003	1	0.44	0.653
Temporal muscle	39.6	0.013	1	0.36	0.607
Medial pterygoid muscle	60.4	0.013	1	0.29	0.601
Lateral pterygoid muscle	57.6	0.019	1	0.29	0.559
Coronoid process	38.0	0.002	1	0.56	0.773
Mandible head	8.2	0.390	1	0.16	0.467

**Table 2 cancers-12-03116-t002:** Dose received by temporomandibular joint-related muscles and bones.

Masseter Muscle	Trismus(+)		Trismus(−)		*p-*Value
	Gy (RBE)	±SD	Gy (RBE)	±SD	
D10	47.7	10.7	36.9	20.9	0.118
D20	35.9	13.4	31.0	18.8	0.278
D30	30.3	15.0	26.4	17.5	0.310
D40	27.2	15.4	22.3	16.8	0.259
D50	24.9	15.2	18.7	16.4	0.204
Temporal muscle					
D10	50.3	14.1	34.2	22.2	0.024
D20	43.4	13.8	31.4	21.8	0.105
D30	37.6	14.7	29.2	21.5	0.187
D40	32.5	16.0	27.2	21.1	0.284
D50	27.9	16.6	25.2	20.6	0.384
Medial pterygoid muscle					
D10	58.3	10.0	44.0	22.7	0.089
D20	51.5	16.6	37.4	24.6	0.117
D30	44.8	21.0	32.3	25.5	0.163
D40	39.4	23.9	27.9	25.5	0.189
D50	35.1	25.8	24.3	25.2	0.201
Lateral pterygoid muscle					
D10	61.2	5.6	46.7	22.1	0.075
D20	57.6	10.9	42.5	22.8	0.077
D30	53.6	15.5	39.1	23.0	0.095
D40	50.1	18.4	36.2	22.8	0.108
D50	46.9	20.5	33.4	22.5	0.116
Coronoid process					
D10	52.2	13.0	29.3	20.5	0.007
D20	50.9	13.3	27.8	20.5	0.007
D30	49.7	13.5	26.7	20.4	0.007
D40	48.6	13.6	25.8	20.2	0.007
D50	47.4	13.7	25.0	20.0	0.008
Mandible head					
D10	25.6	20.2	33.2	25.4	0.253
D20	24.1	19.9	32.1	25.5	0.241
D30	22.8	19.3	31.2	25.6	0.231
D40	21.7	18.5	30.3	25.6	0.222
D50	20.6	17.6	29.5	25.6	0.215

**Table 3 cancers-12-03116-t003:** Patient and tumor characteristics.

Characteristic	
Total (*n*)	31
Follow up time (*m*), mean (range)	57 (6.1–87.1)
Age (y)	59 (31–77)
Gender, *n* (%)	
Male	14 (45)
Female	17 (55)
Primary site, *n* (%)	
Maxillary sinus	8 (26)
Nasal cavity	8 (26)
Parotid gland	5 (16)
Oral cavity	4 (13)
Pharynx	4 (13)
External auditory canal	2 (6)
Histological type, *n* (%)	
Adenoid cystic carcinoma	17 (55)
Olfactory neuroblastoma	5 (16)
Mucoepidermoid carcinoma	4 (13)
Adenocarcinoma	2 (6)
others	3 (10)
Gross tumor volumes (cm^3^)	7.11–129.04
(median: 28.68, average: 39.8)	
Total dose, *n* (%)	
64.0 Gy (RBE)	29 (94)
57.6 Gy (RBE)	2 (6)
T stage, *n* (%)	
T1	0 (0)
T2	5 (16)
T3	7 (23)
T4a	6 (19)
T4b	13 (42)
N stage, *n* (%)	
N0	31 (100)
M stage, *n* (%)	
M0	31 (100)

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
