# Peer review of "Evaluation of Carbon Ion Radiation-Induced Trismus in Head and Neck Tumors Using Dose-Volume Histograms"

_cancers, 2020, doi:10.3390/cancers12113116_

Round 1
Reviewer 1 Report
The article discusses the trismus after Carbon ion radiation therapy in head and neck oncological cases. The idea is interesting and the number of the file sufficient. The methods are sound, and the figures are adequately depicted followed by understandable text. Statistic supports the results. From the clerical point of view no mistakes or discrepancies can be seen within the article. However, the problematics of trismus after the radiation therapy is well known, and the effect of radiation in different types of tissues was also reported. In conclusion I believe that this article might be valuable for the readers, but unlikely it will change the tumours treatment strategies.
Author Response
Response to Reviewer 1 Comments
Reviewer 1
Comments and Suggestions for Authors
The article discusses the trismus after Carbon ion radiation therapy in head and neck oncological cases. The idea is interesting and the number of the file sufficient. The methods are sound, and the figures are adequately depicted followed by understandable text. Statistic supports the results. From the clerical point of view no mistakes or discrepancies can be seen within the article. However, the problematics of trismus after the radiation therapy is well known, and the effect of radiation in different types of tissues was also reported. In conclusion I believe that this article might be valuable for the readers, but unlikely it will change the tumours treatment strategies.
Response: Thank you for reviewing our work and for your constructive feedback. There may be no change in treatment strategy, but it may have some advantages for patients who have previously unnecessarily expanded the area of ​​temporomandibular joint.
Reviewer 2 Report
Thank you for allowing me to review the article entitled Evaluation of carbon ion radiation-induced trismus in head and neck tumors using dose-volume histograms. The article is well presented and the design is correct. I think it is an important and clinically relevant topic. It is suitable for publication
Author Response
Response to Reviewer 2 Comments
Reviewer 2
Comments and Suggestions for Authors
Thank you for allowing me to review the article entitled Evaluation of carbon ion radiation-induced trismus in head and neck tumors using dose-volume histograms. The article is well presented and the design is correct. I think it is an important and clinically relevant topic. It is suitable for publication
Response: Thank you for your encouraging and supportive comments.
Reviewer 3 Report
In the manuscript entitled " Evaluation of carbon ion radiation-induced trismus in head and neck tumors using dose-volume histograms" the Authors assessed the trismus/carbon ion dose relationship using dose-volume histograms in 35 patients who received C-ion RT for head and neck cancer.
The results obtained and presented are new, relevant and significant. The exposition is clear, and the experiments adequately address the hypothesis and support the conclusions.From my point of view the manuscript has important clinical message, and should be of great interest to the readers.
In conclusion, I think that this work meets the high standards of the Journal and I recommend its publication in Cancers with few minor revisions.
Figure 2 represents CT images of a patient with adenoid cystic carcinoma, however this study comprises patients with non-squamous cell carcinoma. Please, solve this discrepancy.
The captation of the figure 1 need to be rephrased.
Author Response
Response to Reviewer 3 Comments
Reviewer 3
Comments and Suggestions for Authors
In the manuscript entitled " Evaluation of carbon ion radiation-induced trismus in head and neck tumors using dose-volume histograms" the Authors assessed the trismus/carbon ion dose relationship using dose-volume histograms in 35 patients who received C-ion RT for head and neck cancer.
The results obtained and presented are new, relevant and significant. The exposition is clear, and the experiments adequately address the hypothesis and support the conclusions.From my point of view the manuscript has important clinical message, and should be of great interest to the readers.
In conclusion, I think that this work meets the high standards of the Journal and I recommend its publication in Cancers with few minor revisions.
Figure 2 represents CT images of a patient with adenoid cystic carcinoma, however this study comprises patients with non-squamous cell carcinoma. Please, solve this discrepancy.
The captation of the figure 1 need to be rephrased.
Response: Thank you for your remarks. We want to clarify that adenoid cystic carcinoma is one of the non-squamous cell carcinomas; therefore, we believe that our figure caption is accurate, and no modifications are required.